# Hyperparameter tuned deep learning-driven medical image analysis for intracranial hemorrhage detection

Naif Almakayeel[1], E. Laxmi Lydia[2], Oleg Razzhivin[3,4,5], S. Rama Sree[6],
Mohammed Altaf Ahmed[7], Bibhuti Bhusan Dash[8], S. P. Siddique Ibrahim[9]*

1 Department of Industrial Engineering, College of Engineering, King Khalid University, Abha, Saudi Arabia, 2 Department of Computer Science and Engineering, Vignan's Institute of Engineering for Women, Visakhapatnam, AP, India, 3 Department of Psychology and Sport Science, Mamun University, Khiva, Uzbekistan, 4 Department of Physical Culture, Khorezm University of Economics, Urgench, Uzbekistan, 5 Department of Theory and Methods of Physical Culture and Life Safety of Elabuga Institute, Kazan Federal University, Kazan, Russia, 6 Department of CSE, Aditya University, Surampalem, India, 7 Department of Computer Engineering, College of Computer Engineering & Sciences, Prince Sattam Bin Abdulaziz University, Al-Kharj, Saudi Arabia, 8 School of Computer Applications, KIIT Deemed to be University, Bhubaneswar, India, 9 School of Computer Science & Engineering (SCOPE), VIT-AP University, Amaravati, Andhra Pradesh, India

* siddique4x@gmail.com

## Abstract

Intracranial haemorrhage (ICH) is a crucial medical emergency that entails prompt assessment and management. Compared to conventional clinical tests, the need for computerized medical assistance for properly recognizing brain haemorrhage from computer tomography (CT) scans is more mandatory. Various deep learning (DL) and artificial intelligence (AI) technologies have been successfully implemented for the analysis of medical images, namely grading of diabetic retinopathy (DR), breast cancer detection, skin cancer detection, and so on. Furthermore, the AI approach ensures accurate detection to facilitate early detection, drastically decreasing the mortality rate. Based on DL models, there are already various techniques for ICH-detection. This manuscript proposes the design of a Hyperparameter Tuned Deep Learning-Driven Medical Image Analysis for Intracranial Hemorrhage Detection (HPDL-MIAIHD) technique. The proposed HPDL-MIAIHD technique investigates the available CT images to classify and identify the ICH. In the presented HPDL-MIAIHD technique, the median filtering (MF) approach is utilized for the image preprocessing step. Next, the HPDL-MIAIHD approach uses an enhanced EfficientNet technique to extract feature vectors. To increase the efficiency of the EfficientNet method, the hyperparameter tuning process is performed by utilizing the chimp optimizer algorithm (COA) method. The ICH detection process is accomplished by the ensemble classification process, comprising long short-term memory (LSTM), stacked auto-encoder (SAE), and bidirectional LSTM (Bi-LSTM) networks. Lastly, the Bayesian optimizer algorithm (BOA) is implemented for the hyperparameter selection of the DL

**Data availability statement:** The dataset implemented in this study is not collected by the authors but is publicly available, as cited in [34] (https://physionet.org/content/ct-ich/1.3.1/). The dataset comprises of 82 CT scans of patients with traumatic brain injury (TBI), with regions of intracranial hemorrhage (ICH) and fractures annotated by radiologists. It encompasses scans from 75 subjects in NIfTI format. This dataset is publicly accessible, and no special access privileges were provided to the authors. The dataset was collected with ethical approval from the research and ethics board in the Iraqi Ministry of Health, Babil Office (approval #1369).

**Funding:** The author(s) received no specific funding for this work.

**Competing interests:** The authors have declared that no competing interests exist.

technique. A comprehensive simulation was conducted on the benchmark CT image dataset to demonstrate the effectiveness of the HPDL-MIAIHD approach in detecting ICH. The performance validation of the HPDL-MIAIHD approach portrayed a superior accuracy value of 99.02% over other existing models.

## 1. Introduction

ICH is potentially a serious neurological emergency which is categorized by vascular rupture with bleeding into the cranial vault [1]. The delays in recognition of ICH transform into interruptions in active management, which leads to significantly avoiding morbidity, cerebral injury, and mortality. The first and best typical imaging modality is non-contrast computed tomography (NCCT), which has a high sensitivity, fast scan collection time, and high specificity for identifying acute haemorrhage [2]. Additionally, CT is also commonly available, which has lower cost and safety and is more easily accessible when compared to magnetic resonance imaging (MRI). Due to the high workload of clinical radiologists, the recognition of CT studies containing ICH may be behind by challenging acute imaging studies [3]. Technical developments have led to many images for investigation. Moreover, today, radiologists must interpret numerous images by comparing them with investigations of similar regions performed a period ago [4]. Since the 1980s, computer-aided diagnosis (CAD) and automated machine learning (ML) methods have been suggested as tools for clinical radiology. ML is a branch of AI that helps train computer approaches to investigate data and perform iterations to learn rules and relationships and improve performance [5].

In recent years, the ML has become more popular. Moreover, DL is a branch of ML that applies multi-layered networks [6]. Medical images have given numerous potential for extracting significant data. In medical fields, DL has been involved in grading recognition in histologic units of lymph nodes, DR on retinal fundus images, and classification of images of skin cancer with improved accuracies [7]. Convolution neural networks (CNN) have been verified to be precise in image processing tasks and classification. In the CT scan image, the same technique will be implemented [8]. The processed image and image segmentation will be trained using a DL method to detect haemorrhage in the scan. It is classified into with and without haemorrhage cases with improved accuracy. Several challenges occur in learning ICH, which affects the automatic systems as a clinical tool [9]. Numerous methods have been available in the literature, characterized by varying datasets through variations in exact preprocessing techniques, implementation methods, DL network architectures and image labelling [10].

This manuscript proposes the design of a Hyperparameter Tuned Deep Learning-Driven Medical Image Analysis for Intracranial Hemorrhage Detection (HPDL-MIAIHD) technique. The proposed HPDL-MIAIHD technique investigates the available CT images to classify and identify the ICH. In the presented HPDL-MIAIHD technique, the median filtering (MF) approach is utilized for the image preprocessing step. Next, the HPDL-MIAIHD approach uses an enhanced EfficientNet technique

to extract feature vectors. To increase the efficiency of the EfficientNet method, the hyperparameter tuning process is performed by utilizing the chimp optimizer algorithm (COA) method. The ICH detection process is accomplished by the ensemble classification process, comprising long short-term memory (LSTM), stacked autoencoder (SAE), and bidirectional LSTM (Bi-LSTM) networks. Lastly, the Bayesian optimizer algorithm (BOA) is implemented for the hyperparameter selection of the DL technique. A comprehensive simulation was conducted on the benchmark CT image dataset to demonstrate the effectiveness of the HPDL-MIAIHD approach in detecting ICH. The key contribution of the HPDL-MIAIHD approach is listed below.

- The HPDL-MIAIHD technique utilizes the MF approach for image preprocessing, effectually mitigating noise and improving image quality. This technique enhances the clarity of CT images, ensuring improved feature extraction. Reducing noise lays a robust foundation for the subsequent steps in ICH detection.

- The enhanced EfficientNet technique is utilized by the HPDL-MIAIHD method for feature extraction, giving a robust and efficient way to capture relevant image features. The COA method is used to fine-tune the model's hyperparameters to optimize the performance of EfficientNet. This approach improves the accuracy and efficiency of feature extraction, improving overall ICH detection performance.

- An ensemble classification process is implemented by incorporating LSTM, SAE, and Bi-LSTM networks, utilizing the merits of each model for ICH detection. This methodology improves robustness by integrating diverse DL techniques. The combination ensures accurate classification and improves the overall reliability of the model in detecting ICH.

- The BOA method is integrated to effectively tune the hyperparameters of the DL techniques used in the HPDL-MIAIHD model. This optimization process improves model performance by choosing the most effective hyperparameters, significantly enhancing the overall accuracy and efficiency of the ICH detection system.

- The HPDL-MIAIHD approach integrates COA-optimized EfficientNet for feature extraction with an ensemble classification model comprising LSTM, SAE, and Bi-LSTM, improving the robustness of ICH detection. Additionally, the use of BOA-driven hyperparameter tuning improves model efficiency and accuracy. The novelty is integrating advanced optimization algorithms, DL models, and hyperparameter tuning to attain superior ICH detection performance.

## 2. Related works

Ragab et al. [11] designed a Political Optimization with DL-based-ICH Diagnosis on Healthcare Management (PODL-ICHDHM) method. Bilateral filtering (BF) was implemented to preprocess the CT images. The Faster SqueezeNet method is employed to extract features. Finally, the PO method with a denoising autoencoder (DAE) approach is used to classify ICH accurately. Nawabi et al. [12] developed a fully automated tool for differentiating neoplastic and non-neoplastic ICH using CT scans, utilizing nnU-Net for segmentation and ResNet-34 for classification. The study specifically addressed neoplastic ICH using advanced models like LeNet, Inception-ResNet, and ResNet-34. In [13], a state-of-the-art DL-based ICH analysis and classification technique was introduced. Subsequently, the DL-based Inceptionv4 model was implemented as a multilayer perceptron (MLP), and a feature extractor could be used for classification. Karthik et al. [14] developed automatic ICH diagnoses and classification using the Rider Optimizer with DL (ICHDC-RODL) approach. This approach produces features by applying the Xtended Central Symmetric Local Binary Pattern (XCS-LBP) framework. Additionally, the Bi-LSTM technique was utilized to detect ICH. Eventually, the ROA was used for the hyperparameter tuning method. In [15], a two-phase automatic technique to diagnose and classify ICH from sinograms employing a DL model was presented. The primary phase is the Intensity Transformed Sinogram Synthesizer, which produces sinograms equivalent to the intensity-transformed CT images. The secondary phase includes a CNN-RNN approach to classify and identify haemorrhages. Venugopal et al. [16] introduced a fusion-based feature extractor technique with a DL framework for ICH Classification and Diagnosis

(FFEDL-ICH). The Gaussian Filtering (GF) and Density-based Fuzzy C-Means (DFCM) methods have been employed for segmentation and preprocessing. This approach could be applied with deep features (ResNet-152) and hand-crafted features (Local Binary Patterns) for extraction. Eventually, a deep neural network (DNN) was employed for classification. In [17], an automatic intracerebral haemorrhage analysis with the help of fusion-based DL with swarm intelligence (AICH-FDLSI) method was developed. The MF and seagull optimizer algorithm (SOA) method is exploited for segmentation and preprocessing. Additionally, EfficientNet and CapsNet are implemented for extraction. Then, a fuzzy support vector machine (FSVM) and deer hunting optimization (DHO) framework were employed for classification and optimization.

Rajagopal et al. [18] implemented a multiple-label ICH classification problem with six haemorrhages. This study introduced a hybrid DL method integrating CNN and LSTM models (Conv-LSTM). Moreover, a Systematic Windowing method with a Conv-LSTM was employed to develop possible solutions. In [19], a fully unsupervised DL technique was presented. This model uses unsupervised principal component analysis (PCA-Net) for removal. In addition, a K-means classifier must train to apply the feature extractor from PCA-Net for spotting ICH. A supervised linear SVM model was also trained using the feature extractor from PCA-Net for a comparison study related to the K-means technique. Aarthy, Muthupriya, and Balaji [20] introduce the gated convolutional neural network (GCNN) method comprising preprocessing, edge detection using scaled region edge detector (SRED), segmentation with screen cluster area segmentation (SCAS), and feature extraction using GCNN with maximum standard external regions (MSER) for improved feature quality and analysis. Choi et al. [21] assess a DL-based ICH detection algorithm (DLHD) in a clinical setting, evaluating diagnostic performance and decision-making consistency among emergency medical professionals. Maiti et al. [22] present a ResNet-10 CNN model with a residual hybrid attention module (RHAM) for classifying brain CT images, improving tumour diagnosis and feature specificity while using a global media collecting layer and dropout mechanism to improve performance. D'Angelo et al. [23] analyze a DL-based pipeline using a Dense-UNet architecture for the assessment of acute ICH on non-contrast computed tomography (NCCT) head scans after traumatic brain injury (TBI). Sindhura and Gorthi [24] propose a DL framework that utilizes synthesized CT images with clinical brain data to enhance haemorrhage detection and segmentation, creating an asymmetry map to highlight differences between the left and right halves of the CT image. Qiao et al. [25] developed a quick and reliable methodology for predicting stroke-associated pneumonia (SAP) in ICH patients utilizing both clinical data and CT scan images for improved accuracy and real-time applicability. Isikbay et al. [26] analyze a DL method for bone removal in NCCT images, evaluating segmentation accuracy and its impact on detecting SDH by junior radiology trainees through a reader study. Ahmed et al. [27] analyze and review DL and ML methods for the prompt and reliable detection and classification of brain haemorrhages using CT images. Emon et al. [28] propose a fully automated uncertainty-guided framework for ICH segmentation in brain CT scans. The framework is trained on a semi-supervised scheme that leverages labelled and unlabeled data. Liu et al. [29] developed a DL technique for identifying and localizing five ICH subtypes in non-contrast head CT scans.

The existing studies show significant progress in using DL and ML techniques for ICH detection, classification, and segmentation. However, various limitations persist: many models lack large, annotated datasets for training and validation, which affects generalizability and performance in different clinical settings. Additionally, the complexity of model architectures, such as hybrid or ensemble techniques, can lead to real-time applicability and interpretability challenges. The dependence on preprocessing techniques like filtering, edge detection, and segmentation may introduce variability in results. Furthermore, most models are not entirely validated across multicenter datasets, limiting their robustness. There is a requirement for more standardized approaches and transparency in model evaluation, specifically in clinical environments, to improve the consistency and trustworthiness of automated systems in ICH diagnosis.

## 3. The proposed model

In this study, an advanced HPDL-MIAIHD technique for automatic and exact ICH detection and identification tasks are proposed. The presented model inspects the available CT images for ICH identification. In the planned HPDL-MIAIHD approach, numerous sub-processes are involved, such as MF-based preprocessing, boosted EfficientNet feature

extractor, COA-based parameter tuning, ensemble classification, and BOA-based parameter optimizer. Fig 1 depicts the complete flow of the HPDL-MIAIHD method.

### 3.1. MF-based preprocessing

In the presented HPDL-MIAIHD technique, the noise is eliminated using the MF approach [30]. This method is chosen due to its efficiency in removing noise while preserving the edges and crucial details of the image. Unlike conventional smoothing techniques, MF replaces each pixel with the median value of its neighbours, which assists in mitigating salt-and-pepper noise without blurring the image. This characteristic makes MF particularly appropriate for medical images, where preserving fine details is significant for accurate diagnosis. Moreover, MF is computationally effective and simple to implement, which ensures quick processing. Its ability to handle diverse types of noise makes it a robust choice over more complex filters, such as Gaussian or Wiener filters, especially when real-time performance is required.

MF is a classical image preprocessing approach that includes replacing pixels in an image with the median value of adjacent pixels within the defined window. This technique successfully reduces noise like salt-and-pepper noise whilst ensuring image and edge details. Mainly, MF is valuable in medical image analysis involving ICH detection, as it improves the clarity and quality of clinical images, which makes succeeding diagnosis and analysis accurate and more reliable.

### 3.2. Feature extraction using improved efficientnet

The improved EfficientNet model is applied to derive a collection of feature vectors [31]. This technique is chosen due to its state-of-the-art performance in image classification tasks, specifically in handling complex medical imaging datasets. EfficientNet achieves superior accuracy while maintaining computational efficiency using a compound scaling method that optimally scales network depth, width, and resolution. This allows for extracting high-quality features without the heavy computational cost typically associated with larger models. Compared to conventional architectures, such as VGG or ResNet, EfficientNet strikes a better balance between accuracy and efficiency, making it ideal for processing medical images where computational resources may be limited. Additionally, its ability to generalize to diverse image types ensures robust feature extraction for different haemorrhage classifications in CT scans.

EfficientNet uses the MBConv as a building block and adopts the squeeze-and-excitation (SE) model to adjust the feature maps. $C_i$ and $C_o$ are both input and output channels, correspondingly. $BN$ indicates batch normalization. $H$ and $W$ represent the height and width of the mapping features. D9WConv is a depthwise convolutional layer, the filter dimensional of which is $K$. Sigmoid and Swish are the two activation functions. During the SE model, $C$ denotes the channel of feature maps, and the parameter refers to the reduction dimension fixed at 4. Firstly, the input channel is increased by 1x1 point-wise convolution (PW), then feature extraction by 3x3 or 5x5 DWConv. The global feature is extracted through global average pooling, and later, channel weight is attained by the sigmoid and two FC layers during the SE model. Lastly, the channel of the feature maps is modified by PW. EfficientNetB0 is an elegant and simple method that integrates the benefits of MobileNet_v2 but with better extraction feature abilities. Fig 2 illustrates the structure of EfficientNet. This study presents an efficient and lightweight waste image classification algorithm using EfficientNetB0, called GECM-EfficientNet. Firstly, several MBConv models are adjusted to decrease their parameters. Next, the efficient channel attention (ECA) model is used to substitute the SE model, which resolves the deficiencies of the reduction dimension. Then, a parallel connection is made between the ECA and the coordinate attention (CA) modules, which enables the spatial weighting function. Lastly, transfer learning is used to initialize the model parameter in the training process.

### 3.3. Hyperparameter tuning using COA

For the hyperparameter tuning process, the COA is used [32]. This model was chosen due to its ability to effectively balance exploration and exploitation in large search spaces, making it appropriate for optimizing DL models. COA replicates the behaviour of chimpanzees in problem-solving, which assists in converging to an optimal solution with fewer iterations

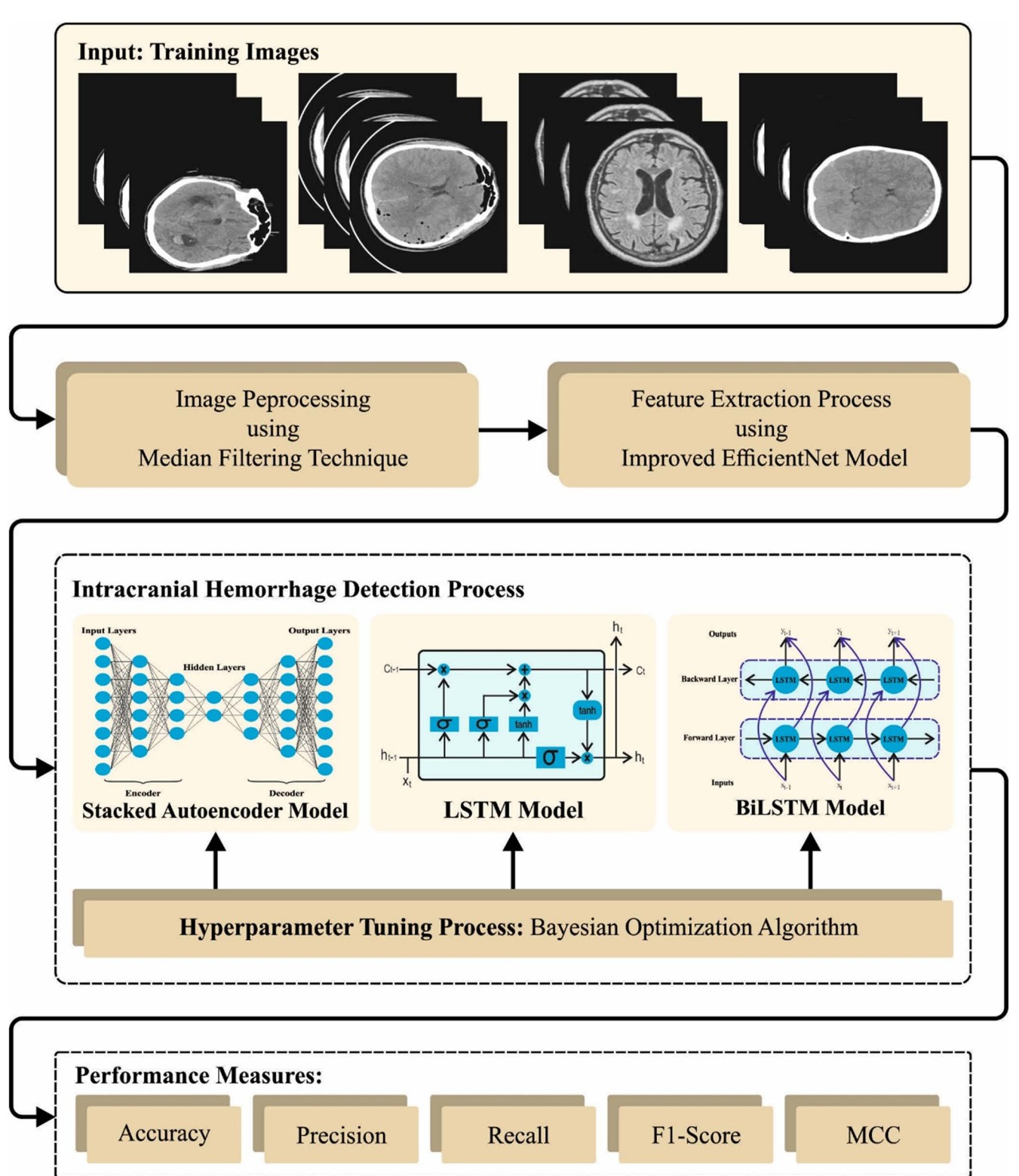

**Fig 1. Overall flow of the HPDL-MIAIHD approach.**

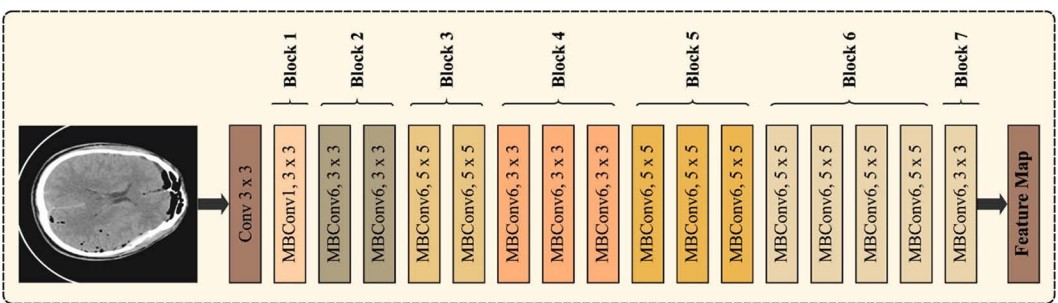

**Fig 2. Architecture of EfficientNet.**

compared to conventional techniques like grid search or random search. Its robustness in escaping local minima and handling complex, nonlinear objective functions improves the model's performance. Additionally, COA's flexibility in adjusting multiple parameters helps fine-tune the model's architecture, improving generalization and efficiency. Compared to other optimization methods, the algorithm's faster convergence and lower computational overhead justify its merit in real-time applications.

COA is a novel meta-heuristic optimization technique based on the social behaviours of chimpanzees in a group. The mathematical modelling of COA is given below.

$$\beta_r = \left| \gamma . x_{prey}(i) - m . x_{chimp}(i) \right| \tag{1}$$

$$x_{chimp}(i+1) = x_{prey}(i) - \alpha . \beta \tag{2}$$

Where $i$ refers to the number of existing iterations, $x_{pery}$ indicates the prey location and $x_{chimp}$ shows the chimp location. $\alpha, m$ and $\gamma$ vectors are evaluated as follows:

$$\alpha = 2.f.r_1 - f \tag{3}$$

$$\gamma = 2.r_2 \tag{4}$$

In the iteration model, the coefficient $f$ is decreased from 2.5 to $0$, $r_1$. and $r_2$ are randomly generated values within [0,1]. The attacking strategy of chimp is defined below:

$$\delta_{attacker} = \left| c_1 . \alpha_{attacker} - m_1 . x \right| \tag{5}$$

$$\delta_{barrier} = \left| c_2 . \alpha_{barrier} - m_2 . r \right| \tag{6}$$

$$\delta_{chaser} = \left| c_3 . \alpha_{chaser} - m_3 . x \right| \tag{7}$$

$$\delta_{diver} = \left| c_4 . \alpha_{diver} - m_4 . x \right| \tag{8}$$

The value of the random vector defines the initial position of the chimp. The next location of the chimp will be between its existing location and prey once the value ranges within$[-1, 1]$. The updating strategy is given in the following:

$$x_n = \alpha_{attacker} - a_n.\delta_{attacker} \tag{9}$$

$$a_2 = \alpha_{barrier} - a_2.\delta_{barrier} \tag{10}$$

$$x_3 = \alpha_{chaser} - a_3.\delta_{chaser} \tag{11}$$

$$x_4 = \alpha_{diver} - a_4.\delta_{diver} \tag{12}$$

$$x_{t_i+1} = \frac{x_1 + x_2 + x_3 + x_4}{4} \tag{13}$$

The subsequent formula is used for updating the chimp location, where $m$ is the chaotic value arbitrarily attained for all the chimps, and the distribution of $m$ is selected based on the optimization problems.

$$\alpha_{chimp}(n_i + 1) = \begin{cases} \alpha_{prey}(n_i) - x, \Delta, if \phi < 0.5 \\ m, if \phi > 0.5 \end{cases} \tag{14}$$

The fitness optimal is a significant aspect of the COA model. Encoding outcome is employed to calculate the best performance of the candidate solution. Here, the value of accuracy is the primary condition employed to develop an FF.

$$Fitness = max(P) \tag{15}$$

$$P = \frac{TP}{TP + FP} \tag{16}$$

Where $TP$ and $FP$ signify the true and false positive values.

### 3.4. Ensemble learning-based classification

This work involves the ensemble of three classification models in the ICH diagnosis process [33]. This methodology is chosen because it incorporates the merits of multiple models to improve prediction accuracy and robustness. By integrating diverse algorithms, such as LSTM, SAE, and Bi-LSTM, ensemble learning reduces the risk of overfitting and biases inherent in single models. This approach is particularly effective for complex medical tasks like haemorrhage detection, where diverse feature patterns and classifications must be considered. Integrating different classifiers allows the system to utilize each unique merit, improving generalization. Compared to individual models, ensemble learning gives a more stable and reliable performance, particularly in scenarios with high data variability and complex patterns. AE is a single hidden layer unsupervised NN model, where the resultant layer is equivalent to the input layer.

AE is achieved using encoding and decoding layers, and its mapping functions are described.

$$h = f_1(x_1) = sf_1(W_1 x_1 + b_1) \tag{17}$$

$$x_2 = f_2(h) = sf_2(W_2 h + b_2) \tag{18}$$

Where $x_1 =$ denotes the input of $AE$; $h =$ indicates the join vector between $x_1$ and $x_2$; $x_2 =$ denotes the input of the $AE$; the nonlinear activation function of $sf_1$ is selected as a sigmoid activation function, and tanh functions; the activation function $sf_2$ of the decoder is either sigmoid function or other function.1 $dl$ refers to the input dimension; $dh$ shows the

 

dimension of the latent parameter vector; $2dr$ denotes the output dimension; $b_1 \in R^{1dl}$ and $b_2 \in R^{2dr}$ are the bias vectors. Three typical DL models are SAE, DBN, and DCNN, a hierarchical DNN architecture encompassed by multilayer AEs.

The architecture of elementary NN comprises input, hidden, and output layers. The output is organized through the activation function and the weight which connects that layer. A new type of NN, RNN, was introduced based on the NN. The main distinction between NNs and RNNs is that RNNs find connection weight between the neurons. However, RNN has the problem of vanishing gradients. Thus, some variants of RNN LSTM have been developed to resolve these problems. Based on RNN, LSTM adds three gating mechanisms to control data transmission and the calculation of outcomes. Forget, input and output gates are the three gating mechanisms.

The forgotten gate is calculated by Eq. (19):

$$f_t = \sigma \left( W_f \cdot [h_{t-1}, x_t] + b_f \right) \tag{19}$$

In Eq. (19), $W_f$ and $b_f$ indicate the weight and bias vector of the forgot gate; $f_t$ represents the vector of the input gate; $[h_{t-1}, x_t]$ shows the two vectors connected into a long vector; $\sigma$ denotes the sigmoid function:

$$W_f \cdot [h_{t-1}, x_t] = [W_f] \cdot \begin{bmatrix} h_{t-1} \\ x_t \end{bmatrix}$$

$$[W_{fh} W_{fx}] \begin{bmatrix} h_{t-1} \\ x_t \end{bmatrix} = W_{fh} h_{t-1} + W_{fx} x_t \tag{20}$$

The following expression calculates the input and output gate:

$$i_t = \sigma \left( W_i \cdot [h_{t-1}, x_t] + b_i \right) \tag{21}$$

$$c_t = f_t \cdot c_{t-1} + i_t \cdot \tan h \left( W_c \cdot [h_{t-1}, x_t] + b_c \right) \tag{22}$$

$$o_t = \sigma \left( W_0 \cdot [h_{t-1}, x_t] + b_0 \right) \tag{23}$$

$$h_t = o_t . \tan h \left( c_t \right) \tag{24}$$

Now $h_t$ indicates the output vector; $W_i$, $W_c$, and $W_o$ represent the weight of the respective gate; $b_i$, $b_c$, and $b_o$ are the bias vectors; $i_t$, $o_t$ and $c_t$ show the vector for input, output gate, and cell activations.

Typical RNN and LSTM frequently ignore future data in timing processing, while Bi-LSTM can exploit future data. The elementary concept of Bi-LSTM is that the forward and backward layers of the training series are 2 LSTM models, correspondingly, and the LSTM network is interconnected to single input and output layers. The output layer obtains prior data for all the points in the input series and also obtains future data for every point through these structures:

$$h_{tr} = H \left( W_1 x_t + W_2 h_{(t-1)r} + b_r \right) \tag{25}$$

$$h_{tl} = H \left( W_1 x_t + W_2 h_{(t-1)l} + b_l \right) \tag{26}$$

$$y_t = W_4 h_{tr} + W_6 h_{tl} + b_y \tag{27}$$

Here, $h_{tr}$, $h_{tl}$, and $y_t$ denote the vectors' forward propagation, backpropagation and output layer correspondingly; $W_1$, $W_2$, $W_3$, $W_4$, $W_5$, and $W_6$ show the respective weight coefficient; $b_r$, $b_l$, $b_y$ denoting the corresponding bias vector.

## 3.5. BOA-based parameter selection

Finally, the BOA was employed to tune the parameters of the DL models. In this research, BOA is exploited as an organized method for global optimization [34]. Unlike other optimization algorithms, which include random, manual, and grid search, this approach is more effective because they are computationally high-cost and time-consuming. The BOA follows the Bayes' rule:

$$p\left(w|D\right) = \frac{p\left(D|w\right)p(w)}{p(D)}$$

(28)

In Eq. (28), $p\left(w|D\right)$ represents the posterior probability distribution, $w$ refers to the hidden value, $p(w)$ denotes the prior probability distribution, and $p\left(D|w\right)$ is the probability. Historical data is applied in Bayes' rule for approximating the posterior probability, representing that the outcome of the prior iteration selects the value for the following iteration. Accordingly, compared to random selection, it achieves a better position faster.

Substituting and acquiring are the two sub-models of BOA. The GP (Gaussian process) is a frequent replacement for modelling the main function applied to evaluate the main function. Generally, GP establishes a previous over function transmitted into subsequent over function after seeing a specific function value.

$$f(z) \sim GP\left(m(z), k\left(z_i, z_j\right)\right)$$

(29)

In Eq. (29), $z$ indicates the functional value with the combination of ($z_i$, and $z_j$) in the input domain. $GP$ is a generalization of Gaussian distribution, $m(z)$ denotes the mean function, and $k\left(z_i, z_j\right)$ shows the covariance function. Now, the covariance function (kernel) defines the connectivity between the parameters in the input domain. This kernel is in charge of the amplitude and smoothness of the GP sample. At the same time, the acquisition function of BOA relies on prior observation and is enhanced with repetition. The acquisition model is used to suggest the following site for iterating according to the outcomes of the replacement model. The mathematical expression of BOA is given below:

$$g^* = argminf(g)$$

(30)

In Eq. (30), the group of hyperparameters which produces the lesser value of the main function is g*, and $f(g)$ refers to the main function to reduce the validation error. $g$ is any value of space G.

## 4. Results and discussion

The simulation outcomes of the HPDL-MIAIHD model are tested on the CT image dataset [35], which encompasses 341 images under Subarachnoid (18 Images), Epidural (171 Images), Intraparenchymal (72 Images), Intraventricular (24 Images), and subdural (56 Images). Table 1 defines the details of the database.

Fig 3 represents the confusion matrices achieved by the HPDL-MIAIHD approach under 60:40 and 70:30 of the TRAP/TESP. The outcome states the effective detection and classification of all five classes.

**Table 1. Details on database.**

| Classes | No. of Images |
| --- | --- |
| Epidural | 171 |
| Intraventricular | 24 |
| Intraparenchymal | 72 |
| Subdural | 56 |
| Subarachnoid | 18 |
| **Total No. of Images** | **341** |

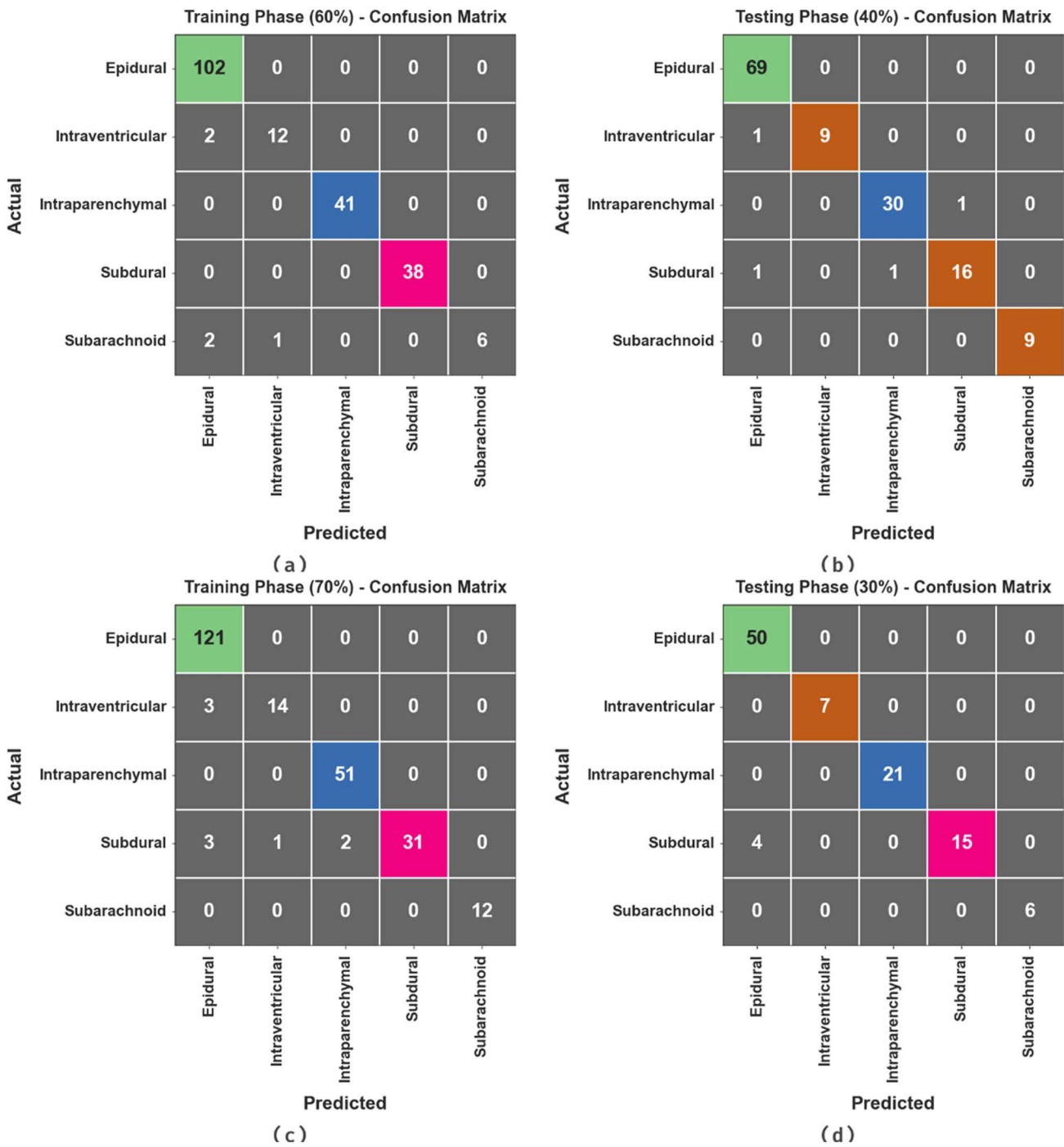

**Fig 3. Confusion matrices of (a-b) 60:40 of TRAP/TESP and (c-d) 70:30 of TRAP/TESP.**

**Table 2. ICH recognition outcome of HPDL-MIAIHD technique on 60:40 of TRAP/TESP.**

| Class | $Accu_y$ | $Prec_n$ | $Reca_l$ | $F_{Score}$ | MCC |
|---|---|---|---|---|---|
| **TRAP (60%)** | | | | | |
| Epidural | 98.04 | 96.23 | 100.00 | 98.08 | 96.15 |
| Intraventricular | 98.53 | 92.31 | 85.71 | 88.89 | 88.17 |
| Intraparenchymal | 100.00 | 100.00 | 100.00 | 100.00 | 100.00 |
| Subdural | 100.00 | 100.00 | 100.00 | 100.00 | 100.00 |
| Subarachnoid | 98.53 | 100.00 | 66.67 | 80.00 | 81.03 |
| **Average** | **99.02** | **97.71** | **90.48** | **93.39** | **93.07** |
| **TESP (40%)** | | | | | |
| Epidural | 98.54 | 97.18 | 100.00 | 98.57 | 97.12 |
| Intraventricular | 99.27 | 100.00 | 90.00 | 94.74 | 94.50 |
| Intraparenchymal | 98.54 | 96.77 | 96.77 | 96.77 | 95.83 |
| Subdural | 97.81 | 94.12 | 88.89 | 91.43 | 90.22 |
| Subarachnoid | 100.00 | 100.00 | 100.00 | 100.00 | 100.00 |
| **Average** | **98.83** | **97.61** | **95.13** | **96.30** | **95.53** |

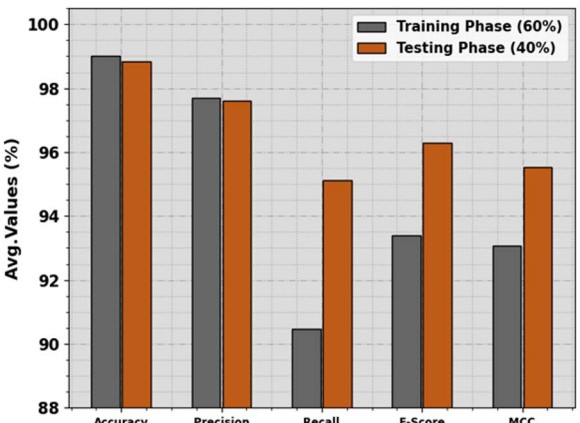

**Fig 4. Average of HPDL-MIAIHD technique on 60:40 of TRAP/TESP.**

Table 2 and Fig 4 state the ICH recognition results of the HPDL-MIAIHD technique below 60:40 of the TRAP/TESP. The outcome implied that the HPDL-MIAIHD technique reaches effectual outcomes under each class. On 60% of the TRAP, the HPDL-MIAIHD technique obtains an average $accu_y$ of 99.02%, $prec_n$ of 97.71%, $reca_l$ of 90.48%, $F_{score}$ of 93.39%, and MCC of 93.07%. Also, on 40% of the TESP, the HPDL-MIAIHD method gains an average $accu_y$ of 98.83%, $prec_n$ of 97.61%, $reca_l$ of 95.13%, $F_{score}$ of 96.30%, and MCC of 95.53%.

Table 3 and Fig 5 state the ICH recognition outcomes of the HPDL-MIAIHD methodology under 70:30 of the TRAP/TESP. The simulation value inferred that the HPDL-MIAIHD methodology gains effective performances under all the classes. On 70% of the TRAP, the HPDL-MIAIHD approach achieves an average $accu_y$ of 98.49%, $prec_n$ of 96.97%, $reca_l$ of 93.23%, $F_{score}$ of 94.87%, and MCC of 93.93%. In addition, on 30% of the TESP, the HPDL-MIAIHD approach attains an average $accu_y$ of 98.45%, $prec_n$ of 98.52%, $reca_l$ of 95.79%, $F_{score}$ of 96.88%, and MCC of 95.87%.

To assess the performance of the HPDL-MIAIHD technique on 60:40 of the TRAP/TESP, TRA $accu_y$ and TES $accu_y$ curves are definite, as exposed in Fig 6. The TRA $accu_y$ and TES $accu_y$ curves reveal the performance of the

**Table 3. ICH recognition outcome of HPDL-MIAIHD methodology on 60:40 of TRAP/TESP.**

| Class | $Accu_y$ | $Prec_n$ | $Reca_l$ | $F_{Score}$ | MCC |
|---|---|---|---|---|---|
| **TRAP (70%)** | | | | | |
| Epidural | 97.48 | 95.28 | 100.00 | 97.58 | 95.07 |
| Intraventricular | 98.32 | 93.33 | 82.35 | 87.50 | 86.80 |
| Intraparenchymal | 99.16 | 96.23 | 100.00 | 98.08 | 97.57 |
| Subdural | 97.48 | 100.00 | 83.78 | 91.18 | 90.20 |
| Subarachnoid | 100.00 | 100.00 | 100.00 | 100.00 | 100.00 |
| **Average** | **98.49** | **96.97** | **93.23** | **94.87** | **93.93** |
| **TESP (30%)** | | | | | |
| Epidural | 96.12 | 92.59 | 100.00 | 96.15 | 92.52 |
| Intraventricular | 100.00 | 100.00 | 100.00 | 100.00 | 100.00 |
| Intraparenchymal | 100.00 | 100.00 | 100.00 | 100.00 | 100.00 |
| Subdural | 96.12 | 100.00 | 78.95 | 88.24 | 86.81 |
| Subarachnoid | 100.00 | 100.00 | 100.00 | 100.00 | 100.00 |
| **Average** | **98.45** | **98.52** | **95.79** | **96.88** | **95.87** |

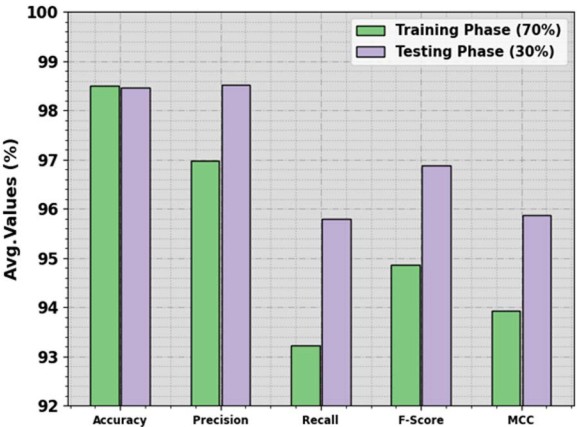

**Fig 5. Average of HPDL-MIAIHD methodology on 70:30 of TRAP/TESP.**

HPDL-MIAIHD technique under many epochs. The outcome provides meaningful facts about the task of learning and generalized abilities of the HPDL-MIAIHD technique. With enhanced epoch counts, the TRA $accu_y$ and TES $accu_y$ curves are perceived to be improved. It is observed that the HPDL-MIAIHD method acquires enhanced testing accuracy that can identify the designs in both data.

Fig 7 exhibits the overall TRA and TES loss values of the HPDL-MIAIHD methodology at 60:40 of the TRAP/TESP under epochs. The TRA loss exhibits the model loss, which is smaller under epochs. The loss rates have been minimal as the model fine-tunes the weight to diminish the prediction fault on both data. The loss curves establish the degree to which the model fits the TRA data. It is observed that the TRA loss and TES loss are progressively diminished, and it is represented that the HPDL-MIAIHD method effactually learns the designs shown in both data. It is also noticed that the HPDL-MIAIHD model modifies the parameters to reduce the discrepancy between the prediction and the original TRA label.

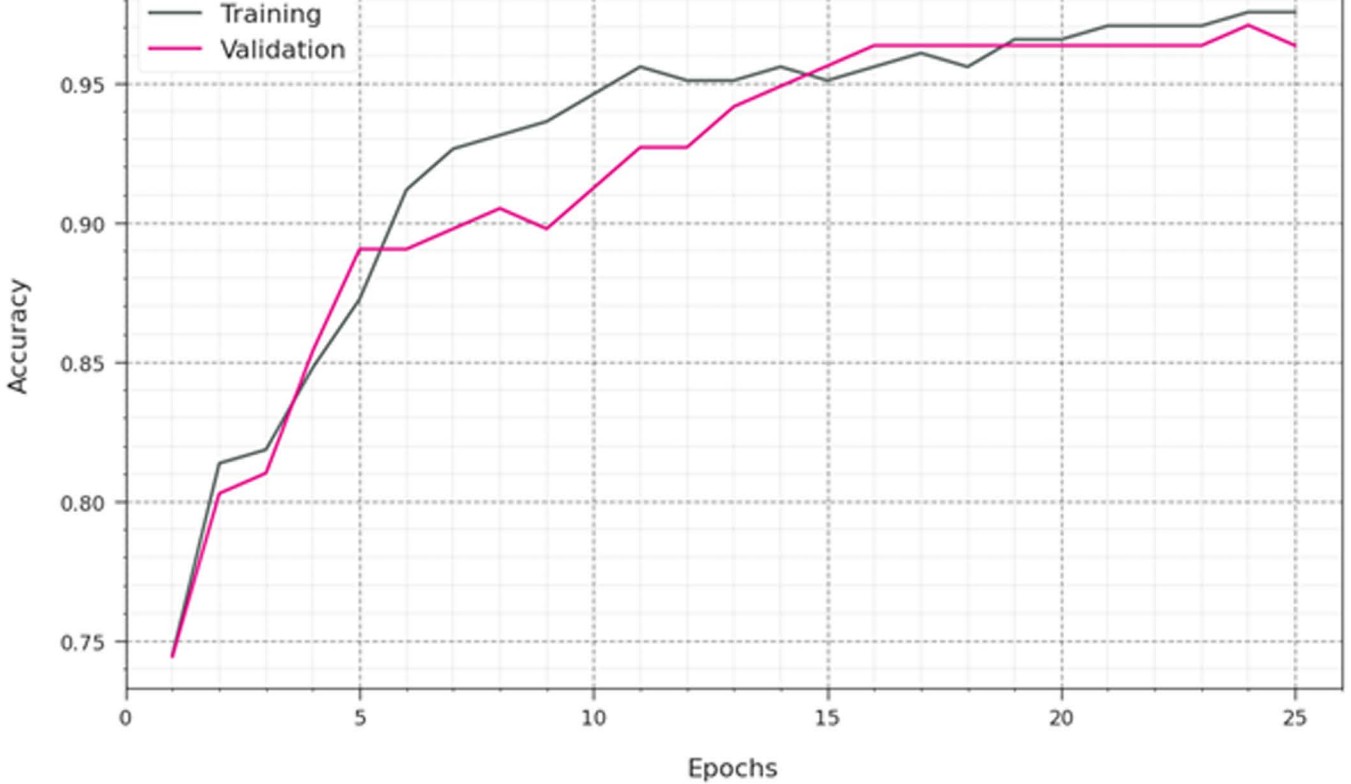

**Fig 6.** $Accu_y$ **curve of HPDL-MIAIHD methodology on 60:40 of TRAP/TESP.**

The PR curve of the HPDL-MIAIHD technique on 60:40 of the TRAP/TESP is established by plotting precision against recall as defined in Fig 8. The outcome confirms that the HPDL-MIAIHD technique gets augmented precision-recall values below five classes. The figure represents the model acquired to identify numerous classes. The HPDL-MIAIHD method attains enhanced performance by detecting positive samples with the least false positives.

Fig 9 epitomizes the ROC curves provided by the HPDL-MIAIHD technique on 60:40 of the TRAP/TESP, which can discriminate the classes. The outcome implies appreciated perceptions of the trade-off among TPR and FPR rates under various classification thresholds and epoch counts. It offers an accurate predictive solution of the HPDL-MIAIHD methodology for classifying dissimilar classes.

Table 4 and Fig 10 demonstrate a detailed examination of the HPDL-MIAIHD technique compared with existing approaches [11–12]. The results exhibited the ineffectual results of the SVM model with a minimal $accu_y$ of 78.24%. Next to that, the DCNN, ResNexT, and UNet models have managed to obtain reasonable performance. Furthermore, the LeNet, Inception-ResNet, and ResNet-34 techniques attain slightly improved outputs. Although the PODL-ICHDHM, AIMA-ICHDC, DL-ICH, and AMG-LSN models certainly attain improved results, the HPDL-MIAIHD technique shows promising outcomes with a maximum $accu_{racy}$ of 99.02%. This result illustrates the robust potential of the model for achieving high classification performance and robustness compared to other existing techniques and its ability to manage complex data and optimize performance confirms its efficiency in real-world applications.

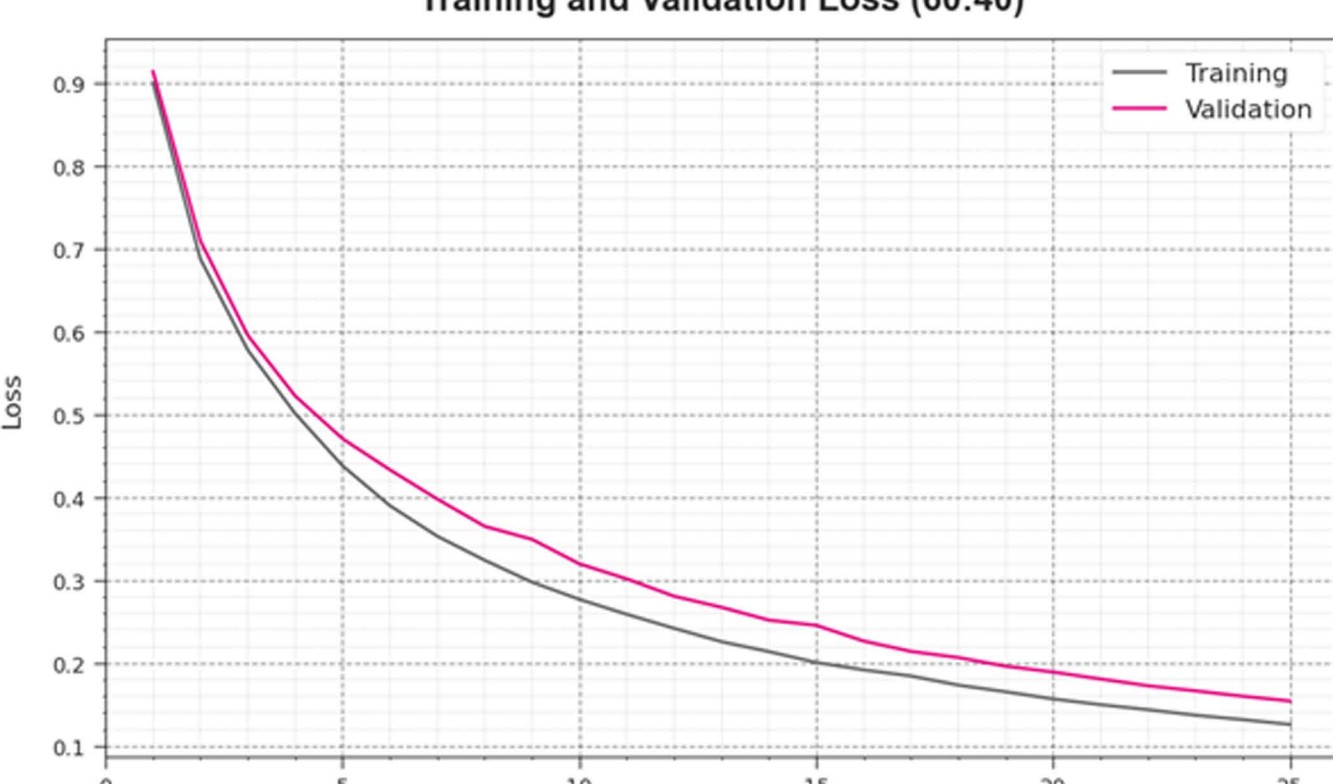

**Fig 7. Loss curve of HPDL-MIAIHD method on 60:40 of TRAP/TESP.**

Table 5 and Fig 11 compare the HPDL-MIAIHD technique regarding computation time (CT). The results showed that the HPDL-MIAIHD technique performed significantly with a minimal CT of 10.62ms. On the other hand, the different existing PODL-ICHDHM, AIMA-ICHDC, DL-ICH, AMG-LSN, DCNN, SVM, ResNexT, and UNet approaches have stated maximal CT values. Moreover, LeNet, Inception-ResNet, and ResNet-34 approaches attained slightly reduced CT values. These results guaranteed the superior solution of the HPDL-MIAIHD approach to the ICH detection process.

## 5. Conclusion

In this article, an innovative HPDL-MIAIHD method for automatic and correct ICH detection and classification tasks are proposed. The proposed HPDL-MIAIHD technique investigates the available CT images to classify and detect the ICH. In the presented HPDL-MIAIHD technique, various sub-processes are involved: MF-based preprocessing, improved EfficientNet feature extraction, COA-based parameter tuning, ensemble classification, and BOA-based parameter optimization. Meanwhile, the HPDL-MIAIHD technique applies an improved EfficientNet model to extract feature vectors. The COA performs the hyperparameter tuning procedure to enhance the efficiency of the improved EfficientNet model. The ICH detection process is carried out by the ensemble classification process, comprising SAE, LSTM, and BiLSTM models. Lastly, the BOA is used for the hyperparameter range of the DL models. A comprehensive simulation was conducted on the benchmark CT image dataset to demonstrate the effectiveness of the HPDL-MIAIHD approach in detecting ICH. The

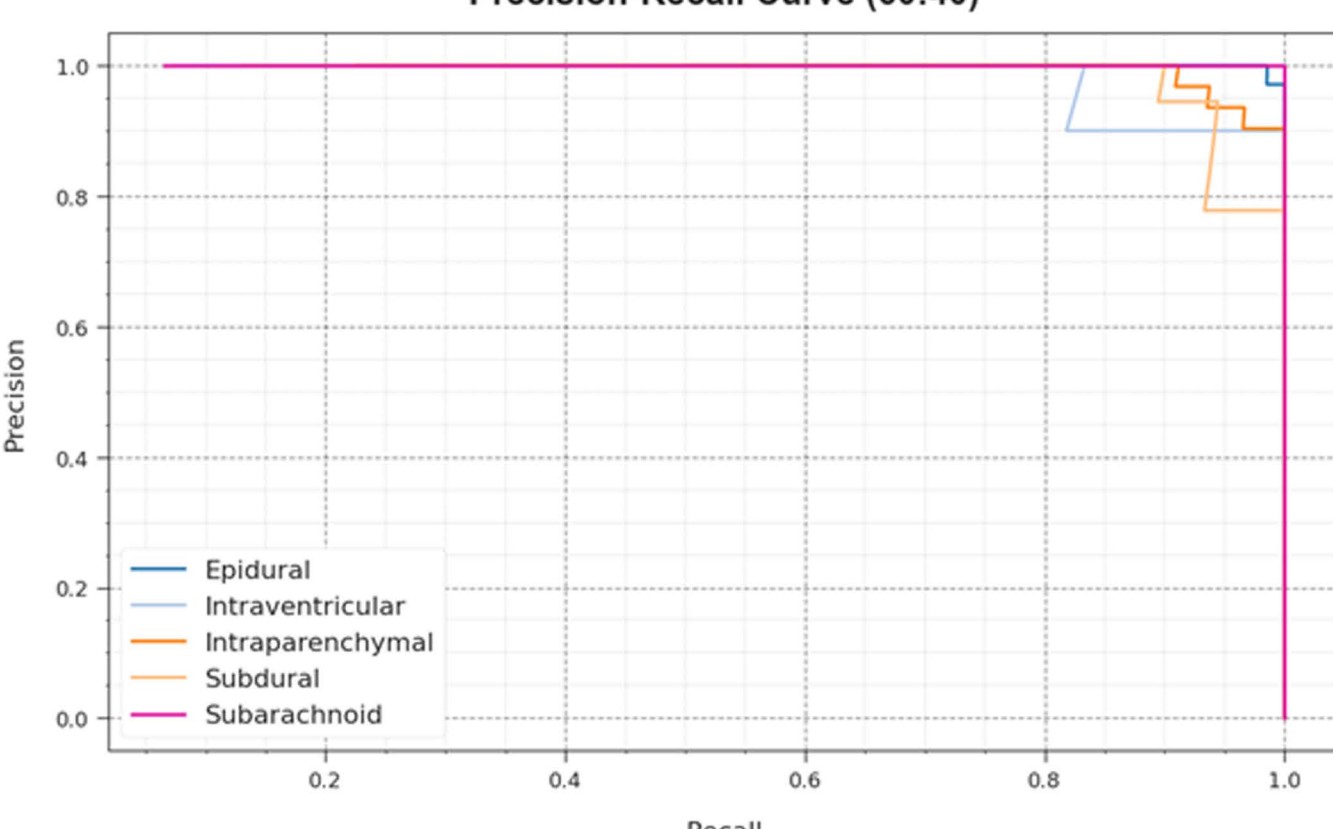

**Fig 8. PR curve of HPDL-MIAIHD method on 60:40 of TRAP/TESP.**

performance validation of the HPDL-MIAIHD approach portrayed a superior accuracy value of 99.02% over other existing models.

### 5.1. Limitations and future directions for enhancing model generalizability and clinical applicability

The HPDL-MIAIHD approach demonstrates high accuracy of 99.02% in detecting ICH on the benchmark CT image dataset; however, its efficiency is restricted by the reliance on a single, limited dataset. This limitation may affect the capability of the model to generalize across a wide range of real-world clinical cases, potentially affecting its performance in diverse patient populations. Moreover, the study concentrates on CT imaging, which is just one diagnostic modality, and does not integrate other imaging techniques usually utilized in clinical practice, such as MRI or ultrasound, which could offer complementary insights. Furthermore, the computational complexity of DL methods poses challenges for real-time application in clinical environments, specifically those with limited resources.

To address these limitations, future work should concentrate on expanding the dataset to comprise more diverse and comprehensive samples from multiple healthcare centers, improving the robustness and generalizability of the model. Integrating multimodal data from diverse imaging techniques, such as MRI or ultrasound, could improve diagnostic accuracy and give a more holistic approach to ICH detection. Additionally, optimizing the computational efficiency of DL techniques is significant to ensure their practicality for real-time clinical use, particularly in resource-constrained settings.

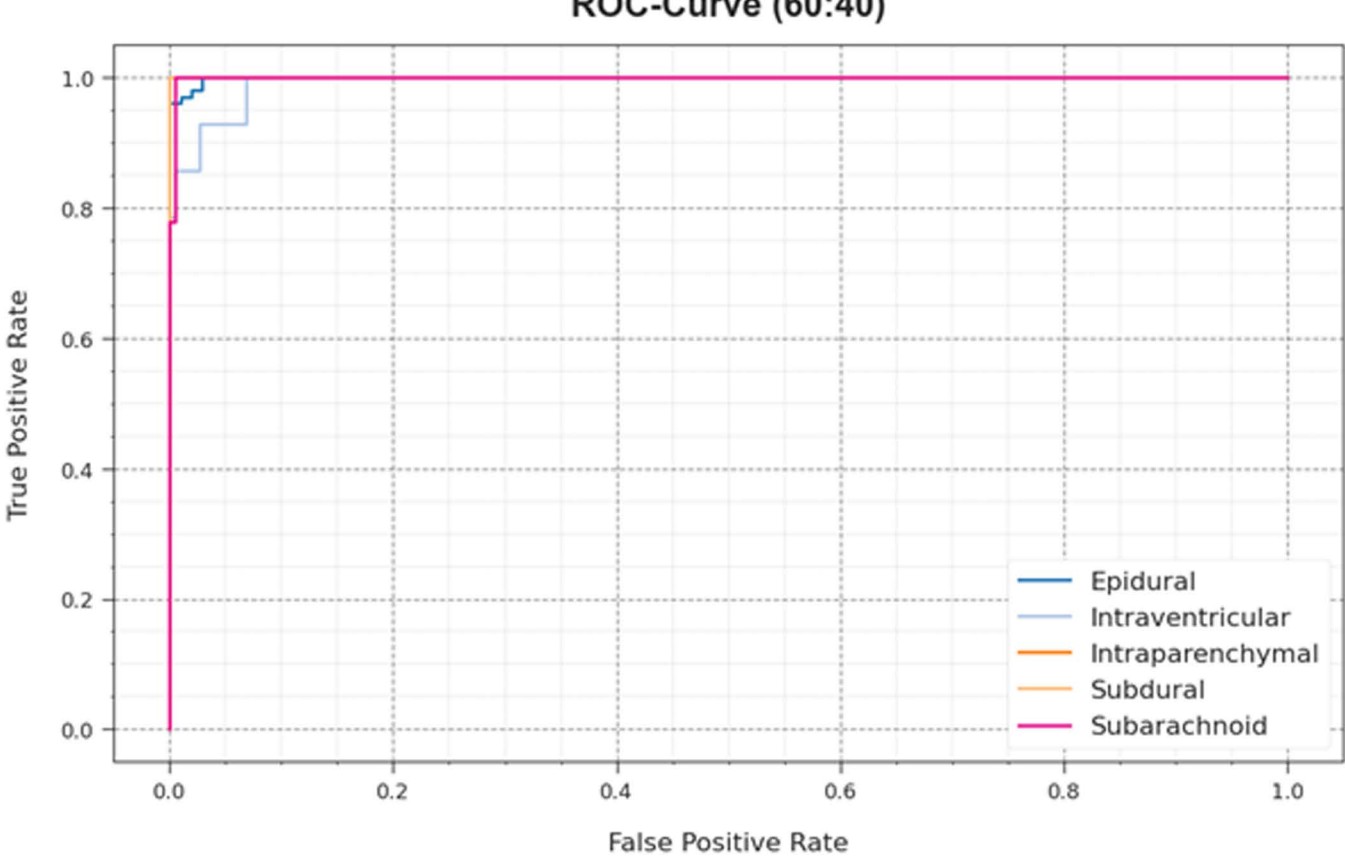

**Fig 9. ROC curve of HPDL-MIAIHD method on 60:40 of TRAP/TESP.**

**Table 4. *Accu_y* outcome of HPDL-MIAIHD technique with recent methods [11–12].**

| Methods | Accuracy (%) |
|---|---|
| HPDL-MIAIHD | 99.02 |
| LeNet | 95.12 |
| Inception-ResNet | 94.12 |
| ResNet-34 | 97.00 |
| PODL-ICHDHM | 98.43 |
| AIMA-ICHDC | 97.13 |
| DL-ICH | 95.64 |
| AMG-LSN | 93.41 |
| DCNN | 87.86 |
| SVM | 78.24 |
| ResNexT | 89.19 |
| U-Net | 88.00 |

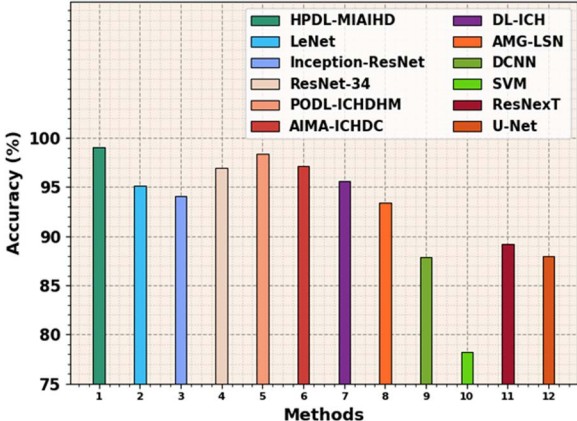

**Fig 10.** $Accu_y$ **outcome of HPDL-MIAIHD technique with recent methods.**

**Table 5. CT outcome of HPDL-MIAIHD technique with recent methods.**

| Methods | CT (ms) |
|---|---|
| HPDL-MIAIHD | 10.62 |
| LeNet | 11.90 |
| Inception-ResNet | 15.16 |
| ResNet-34 | 17.18 |
| PODL-ICHDHM | 31.24 |
| AIMA-ICHDC | 43.63 |
| DL-ICH | 74.47 |
| AMG-LSN | 42.22 |
| DCNN | 35.56 |
| SVM | 53.05 |
| ResNeT | 57.27 |
| UNet | 62.52 |

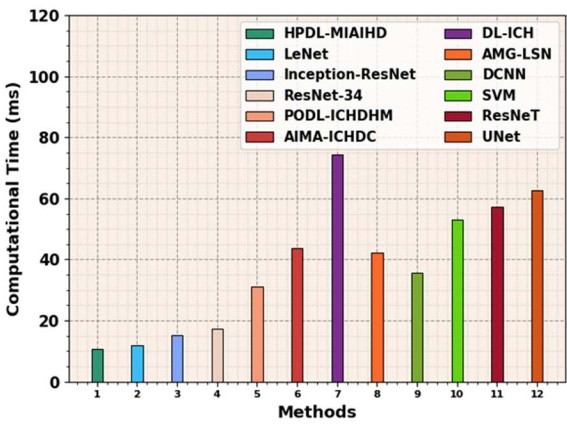

**Fig 11. CT outcome of HPDL-MIAIHD technique with other methods.**

Lastly, developing standardized evaluation protocols and ensuring seamless integration with clinical decision support systems would significantly enhance the usability and effectualness of the methodology in everyday medical practice.

## Author contributions

**Conceptualization:** Naif Almakayeel, E. Laxmi Lydia, S. Rama Sree, Mohammed Altaf Ahmed.

**Data curation:** Bibhuti Bhusan Dash.

**Formal analysis:** Oleg Razzhivin.

**Methodology:** S. Rama Sree.

**Validation:** S.P. Siddique Ibrahim.

**Writing – original draft:** S.P. Siddique Ibrahim.

**Writing – review & editing:** S.P. Siddique Ibrahim.

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
