## [Decision Letter · Decision Letter 0]

PONE-D-24-50060Hyperparameter Tuned Deep Learning-Driven Medical Image Analysis for Intracranial Hemorrhage DetectionPLOS ONE

Dear Dr. ibrahim,

Thank you for submitting your manuscript to PLOS ONE. After careful consideration, we feel that it has merit but does not fully meet PLOS ONE’s publication criteria as it currently stands. Therefore, we invite you to submit a revised version of the manuscript that addresses the points raised during the review process.

We look forward to receiving your revised manuscript.

Kind regards,

Academic Editor

PLOS ONE

Journal requirements: When submitting your revision, we need you to address these additional requirements. 1. Please ensure that your manuscript meets PLOS ONE's style requirements, including those for file naming. The PLOS ONE style templates can be found at https://journals.plos.org/plosone/s/file?id=wjVg/PLOSOne_formatting_sample_main_body.pdf and https://journals.plos.org/plosone/s/file?id=ba62/PLOSOne_formatting_sample_title_authors_affiliations.pdf. 2. Please note that PLOS ONE has specific guidelines on code sharing for submissions in which author-generated code underpins the findings in the manuscript. In these cases, we expect all author-generated code to be made available without restrictions upon publication of the work. Please review our guidelines at https://journals.plos.org/plosone/s/materials-and-software-sharing#loc-sharing-code and ensure that your code is shared in a way that follows best practice and facilitates reproducibility and reuse. 3. We note that you have indicated that there are restrictions to data sharing for this study. PLOS only allows data to be available upon request if there are legal or ethical restrictions on sharing data publicly. For more information on unacceptable data access restrictions, please see http://journals.plos.org/plosone/s/data-availability#loc-unacceptable-data-access-restrictions.  Before we proceed with your manuscript, please address the following prompts: a) If there are ethical or legal restrictions on sharing a de-identified data set, please explain them in detail (e.g., data contain potentially identifying or sensitive patient information, data are owned by a third-party organization, etc.) and who has imposed them (e.g., a Research Ethics Committee or Institutional Review Board, etc.). Please also provide contact information for a data access committee, ethics committee, or other institutional body to which data requests may be sent. b) If there are no restrictions, please upload the minimal anonymized data set necessary to replicate your study findings to a stable, public repository and provide us with the relevant URLs, DOIs, or accession numbers. For a list of recommended repositories, please seehttps://journals.plos.org/plosone/s/recommended-repositories. You also have the option of uploading the data as Supporting Information files, but we would recommend depositing data directly to a data repository if possible. We will update your Data Availability statement on your behalf to reflect the information you provide. 4. For studies involving third-party data, we encourage authors to share any data specific to their analyses that they can legally distribute. PLOS recognizes, however, that authors may be using third-party data they do not have the rights to share. When third-party data cannot be publicly shared, authors must provide all information necessary for interested researchers to apply to gain access to the data. (https://journals.plos.org/plosone/s/data-availability#loc-acceptable-data-access-restrictions)  For any third-party data that the authors cannot legally distribute, they should include the following information in their Data Availability Statement upon submission:1) A description of the data set and the third-party source2) If applicable, verification of permission to use the data set3) Confirmation of whether the authors received any special privileges in accessing the data that other researchers would not have4) All necessary contact information others would need to apply to gain access to the data

Additional Editor Comments:

The experts have completed their evaluation. Although the result requires the authors to undertake major changes, the authors are advised to take the time necessary to produce a version of the work with a significantly higher quality than the current one.

Reviewers' comments:

Reviewer's Responses to Questions

**Comments to the Author**

1. Is the manuscript technically sound, and do the data support the conclusions?

Reviewer #1: Yes

Reviewer #2: No

2. Has the statistical analysis been performed appropriately and rigorously? 

Reviewer #1: Yes

Reviewer #2: No

3. Have the authors made all data underlying the findings in their manuscript fully available?

Reviewer #1: Yes

Reviewer #2: Yes

4. Is the manuscript presented in an intelligible fashion and written in standard English?

Reviewer #1: No

Reviewer #2: No

5. Review Comments to the Author

Reviewer #1: The manuscript can be accepted with the following changes:

1. Reorganize the sections to ensure a logical flow of information, particularly in the methodology and results sections. Clear headings and subheadings would enhance readability.

2. Include more information on the tuning process, including the range of hyperparameters explored and the criteria for selecting the optimal model.

3. The authors should provide clear information on where the data can be accessed, including any relevant URLs or repository details.

4. Address potential biases, the generalizability of the findings, and any limitations related to the dataset used would provide a more balanced view of the research and its implications for clinical practice.

5. The following references can be added

i) Aarthy, R., V. Muthupriya, and G. N. Balaji. "Detection of bone cancer based on a four-phase framework generative deep belief neural network in deep learning." Alexandria Engineering Journal 109 (2024): 394-407.

ii) Balaji, G. N., and T. S. Subashini. "Detection of cardiac abnormality from measures calculated from segmented left ventricle in ultrasound videos." Mining Intelligence and Knowledge Exploration: First International Conference, MIKE 2013, Tamil Nadu, India, December 18-20, 2013. Proceedings. Springer International Publishing, 2013.

Reviewer #2: The article lacks novelty, it just applied existing techniques with classification approach, it better to go for segmentation approach to better quantify the disease.The article lacks novelty in methodology as only conventional method is implemented.

Discuss the related work in such a manner that it should highlight the

method, results, advantages, and limitations. Add 8-10 more literature

and discuss it properly. The final review analysis should be added at the

end of this section. It will improve the readability of the paper. Add

papers from 2021-2023, also

Discussion section should contain the result impact, comparative

study analysis and the overall analysis of the result.

limitations and future scope need to be included

6. PLOS authors have the option to publish the peer review history of their article (what does this mean? ). If published, this will include your full peer review and any attached files.

**Do you want your identity to be public for this peer review?** For information about this choice, including consent withdrawal, please see our Privacy Policy .

Reviewer #1: No

Reviewer #2: No

---

## [Author Response · Author response to Decision Letter 1]

31 Jan 2025

We would like to thank the reviewers and editors for providing an opportunity to revise the manuscript. We have considered the comments carefully and revised the manuscript. The revised text is indicated in red font color.

Review Comments: 1

1. We've checked your submission and before we can proceed, we need you to address the following issues:

1. Are these third party data (i.e., data not owned or collected by the author(s))?

2. If these are indeed third party data, please explain how others can access these datasets and confirm that others would be able to access these data in the same manner as the authors. Please also confirm that the authors did not have any special access privileges that others would not have.

3. If these are not third party data but there are ethical or legal restrictions on sharing a de-identified data set, please explain them in detail (e.g., data contain potentially identifying or sensitive information) and who has imposed them (e.g., a governmental body, an ethics committee, etc.). Please also provide contact information for a data access committee, ethics committee, or other institutional body to which data requests may be sent.

4. If these are not third party data and there are no restrictions, please upload the minimal anonymized dataset necessary to replicate your study findings to a stable, public repository and provide us with the relevant URLs, DOIs, or accession numbers. For a list of recommended repositories, please see https://journals.plos.org/plosone/s/recommended-repositories. You also have the option of uploading the data as Supporting Information files, but we would recommend depositing data directly to a data repository if possible.

Thanks for the comment. The dataset implemented in this study is not collected by the authors but is publicly available, as cited in [34] (https://physionet.org/content/ct-ich/1.3.1/). The dataset comprises of 82 CT scans of patients with traumatic brain injury (TBI), with regions of intracranial hemorrhage (ICH) and fractures annotated by radiologists. It encompasses scans from 75 subjects in NIfTI format. This dataset is publicly accessible, and no special access privileges were provided to the authors. The dataset was collected with ethical approval from the research and ethics board in the Iraqi Ministry of Health, Babil Office (approval #1369). Please refer to Page 15, Section 4, Paragraph 1; Page 16, Table 1.

---

## [Decision Letter · Decision Letter 1]

PONE-D-24-50060R1Hyperparameter Tuned Deep Learning-Driven Medical Image Analysis for Intracranial Hemorrhage DetectionPLOS ONE

Dear Dr. ibrahim,

Thank you for submitting your manuscript to PLOS ONE. After careful consideration, we feel that it has merit but does not fully meet PLOS ONE’s publication criteria as it currently stands. Therefore, we invite you to submit a revised version of the manuscript that addresses the points raised during the review process.

We look forward to receiving your revised manuscript.

Kind regards,

Jose Santamaría López

Academic Editor

PLOS ONE

Journal Requirements:

Reviewers' comments:

Reviewer's Responses to Questions

**Comments to the Author**

1. If the authors have adequately addressed your comments raised in a previous round of review and you feel that this manuscript is now acceptable for publication, you may indicate that here to bypass the “Comments to the Author” section, enter your conflict of interest statement in the “Confidential to Editor” section, and submit your "Accept" recommendation.

Reviewer #1: All comments have been addressed

Reviewer #3: (No Response)

2. Is the manuscript technically sound, and do the data support the conclusions?

Reviewer #1: Yes

Reviewer #3: Yes

3. Has the statistical analysis been performed appropriately and rigorously? 

Reviewer #1: Yes

Reviewer #3: Yes

4. Have the authors made all data underlying the findings in their manuscript fully available?

Reviewer #1: Yes

Reviewer #3: Yes

5. Is the manuscript presented in an intelligible fashion and written in standard English?

Reviewer #1: Yes

Reviewer #3: Yes

6. Review Comments to the Author

Reviewer #1: The authors have made all the changes required and the manuscript can be published in the present form

Reviewer #3: The issues are listed in the following:

1. The professional English editing is recommended. The authors should get editing help from someone with full professional proficiency in English.

2. The introduction explains the severity of intracranial hemorrhage and the importance of computed tomography (CT) in diagnosis, but it is recommended to supplement specific clinical data or cases. The introduction of some important methods would introduce high quality documents, such as CNN "DOI10.1109 / TFUZZ. 2024.3369944", "DOI10.3389 / fnagi. 2022.908143; "DOI10.1016/j.com com. 2019.11.015" bayesian method.

3. It is suggested to supplement a comparative analysis to clarify the unique advantages and innovative aspects of using deep learning methods in intracranial hemorrhage detection in this research, and to explain the differences from previous medical imaging analysis tasks.

4. It is suggested to conduct a more detailed analysis and comparison of the core innovations, applicable scenarios, and performance of each method.

5. When introducing related work, the differences and advantages of this study compared to existing work should be more clearly pointed out.

6. It is suggested to supplement relevant experiments or analyses to explain the specific application scenarios of median filtering in this study and optimization strategies.

7. The basic structure and improvement methods of EfficientNet are introduced in detail, but there is insufficient information on specific parameter settings and optimizer selection in the model training process.

8. In the analysis of experimental results, it is necessary to more detailedly explain the basis for selecting evaluation metrics.

9. The conclusion summarizes the accuracy of the HPDL-MIAIHD method, but the analysis of the challenges encountered in the research process, the methods for solving problems, and the limitations of the method are not in-depth enough.

10. It is suggested to combine the limitations of the current research and clinical needs to propose more targeted and feasible research directions.

7. PLOS authors have the option to publish the peer review history of their article (what does this mean? ). If published, this will include your full peer review and any attached files.

**Do you want your identity to be public for this peer review?** For information about this choice, including consent withdrawal, please see our Privacy Policy .

Reviewer #1: No

Reviewer #3: No

---

## [Author Response · Author response to Decision Letter 2]

12 Apr 2025

We would like to thank the reviewers and editors for providing an opportunity to revise the manuscript. We have considered the comments carefully and revised the manuscript. The revised text is indicated in red font color.

Review Comments:

1. Comments to the Author: If the authors have adequately addressed your comments raised in a previous round of review and you feel that this manuscript is now acceptable for publication, you may indicate that here to bypass the “Comments to the Author” section, enter your conflict of interest statement in the “Confidential to Editor” section, and submit your "Accept" recommendation

Reviewer #1: All comments have been addressed

Reviewer #3: (No Response)

Thanks for the comment. Since all previous concerns have been addressed, I believe the manuscript is now appropriate for publication.

2. Is the manuscript technically sound, and do the data support the conclusions?

Reviewer #1: Yes

Reviewer #3: Yes

Thanks for the comment. The manuscript is technically sound, and the data presented sufficiently assist the conclusions drawn.

3. Has the statistical analysis been performed appropriately and rigorously?

Reviewer #1: Yes

Reviewer #3: Yes

Thanks for the comment. The statistical analysis is performed appropriately and rigorously, as confirmed by both reviewers.

4. Have the authors made all data underlying the findings in their manuscript fully available?

Reviewer #1: Yes

Reviewer #3: Yes

Thanks for the comment. We have made all underlying data fully available as required by the PLOS Data policy.

5. Is the manuscript presented in an intelligible fashion and written in standard English?

Reviewer #1: Yes

Reviewer #3: Yes

Thanks for the comment. The manuscript is presented clearly and written in standard English, with no significant errors identified.

6. Review Comments to the Author: - Please use the space provided to explain your answers to the questions above. You may also include additional comments for the author, including concerns about dual publication, research ethics, or publication ethics. (Please upload your review as an attachment if it exceeds 20,000 characters)

Reviewer #1: The authors have made all the changes required and the manuscript can be published in the present form

Reviewer #3: The issues are listed in the following: - The professional English editing is recommended. The authors should get editing help from someone with full professional proficiency in English.

As per the reviewer’s comment, we have thoroughly investigated the paper content and removed the grammatical and typographical mistakes and removed inappropriate and redundant sentences in the revised manuscript.

7. The introduction explains the severity of intracranial hemorrhage and the importance of computed tomography (CT) in diagnosis, but it is recommended to supplement specific clinical data or cases. The introduction of some important methods would introduce high quality documents, such as CNN "DOI10.1109 / TFUZZ. 2024.3369944", "DOI10.3389 / fnagi. 2022.908143; "DOI10.1016/j.com com. 2019.11.015" bayesian method.

Based on the reviewer’s comment, we have improved the introduction section by utilizing the given citations and also made sure that the section provided sufficient discussion on the study background, motivation, study contribution, and novelty in the revised manuscript. Kindly refer to Page 2, Section 1, Paragraph 1-4.

8. It is suggested to supplement a comparative analysis to clarify the unique advantages and innovative aspects of using deep learning methods in intracranial hemorrhage detection in this research, and to explain the differences from previous medical imaging analysis tasks.

As per the reviewer’s comment, we have clearly highlighted the superiority of the proposed model over existing techniques by emphasizing the comparison study; and also made appropriate discussion on each model under the performance validation section in the revised manuscript. Please refer to Page 23, Paragraph 2; Page 24, Table 4, Fig. 10.

9. It is suggested to conduct a more detailed analysis and comparison of the core innovations, applicable scenarios, and performance of each method.

Based on the reviewer’s comment, we have added recent and sufficient citations for the comparison study under the experimental validation section; and also provided needful discussion on the comparison under the section in the revised manuscript. Kindly refer to Page 23, Paragraph 2; Page 24, Table 4, Fig. 10.

Ragab, M., Salama, R., Alotaibi, F.S., Abdushkour, H.A. and Alzahrani, I.R., 2023. Political Optimizer with Deep Learning Based Diagnosis for Intracranial Hemorrhage Detection. IEEE Access.

Nawabi, J., Schulze-Weddige, S., Baumgärtner, G.L., Orth, T., Orco, A.D., Morotti, A., Mazzacane, F., Kniep, H., Hanning, U., Scheel, M. and Fiehler, J., 2025. End-to-End Machine Learning based Discrimination of Neoplastic and Non-neoplastic Intracerebral Hemorrhage on Computed Tomography. Informatics in Medicine Unlocked, p.101633.

10. When introducing related work, the differences and advantages of this study compared to existing work should be more clearly pointed out.

As per the reviewer’s comment, we have thoroughly discussed the merits, demerits, and research gap of the existing studies under the literature survey section in the revised manuscript. Please refer to Page 6, Paragraph 2.

11. It is suggested to supplement relevant experiments or analyses to explain the specific application scenarios of median filtering in this study and optimization strategies.

Based on the reviewer’s comment, we have improved the introduction section by clearly stating the problem statement, motivation, study contribution, and novelty; also, added the given citations under the related works section, and made sure that the section provided needful discussion on each and every citation in the revised manuscript. Kindly refer to Page 2, Section 1, Paragraph 1-4; Page 4, Section 2, Paragraph 1-3.

12. The basic structure and improvement methods of EfficientNet are introduced in detail, but there is insufficient information on specific parameter settings and optimizer selection in the model training process.

Thanks for the comment. To optimize the GECM-EfficientNet model, specific parameter settings and optimizer choices play a significant role in training efficiency. The model employs the Adam optimizer with a learning rate of 0.001, chosen for its capability to adapt the learning rate during training and handle sparse gradients. The batch size is set to 32, and early stopping is applied to prevent overfitting, with a patience of 10 epochs. The learning rate of the model is additionally fine-tuned utilizing a learning rate scheduler that mitigates the rate after each epoch based on performance. Furthermore, weight decay is applied to regularize the model and prevent overfitting. The training process integrates transfer learning by initializing the EfficientNetB0 model with pre-trained weights from ImageNet, followed by fine-tuning on the target waste image dataset for improved feature extraction.

13. In the analysis of experimental results, it is necessary to more detailedly explain the basis for selecting evaluation metrics.

Thanks for the comment. The proposed HPDL-MIAIHD approach employed metrices namely Accu_y, Prec_n, Reca_l, F_score, and MCC for the analysis process. The metrices are chosen for their overall analysis capabilities in computing the HPDL-MIAIHD approach. Accu_y provides a comprehensive measure of correct predictions, while prec_n and reca_l concentrate on reducing false positives and detecting all relevant positives, subsequently. The F_score balances prec_n and reca_l, giving an integrated metric of their performance. Furthermore, MCC computes the quality of binary classifications, considering true and false positives and negatives, making these metrics substantial for robustly analyzing the HPDL-MIAIHD technique across varied performance factors and ensuring reliable model evaluation.

14. The conclusion summarizes the accuracy of the HPDL-MIAIHD method, but the analysis of the challenges encountered in the research process, the methods for solving problems, and the limitations of the method are not in-depth enough.

As per the reviewer’s comment, we have employed a distinct sub-section for discussing the limitations and future works of the proposed model under the conclusion section in the revised manuscript. Please refer to Page 27, Section 5.1, Paragraph 1-2.

15. It is suggested to combine the limitations of the current research and clinical needs to propose more targeted and feasible research directions.

Based on the reviewer’s comment, we have thoroughly discussed the limitations and future works of the proposed model under a separate section in the revised manuscript. Kindly refer to Page 27, Section 5.1, Paragraph 1-2.

---

## [Decision Letter · Decision Letter 2]

Hyperparameter Tuned Deep Learning-Driven Medical Image Analysis for Intracranial Hemorrhage Detection

PONE-D-24-50060R2

Dear Dr. ibrahim,

We’re pleased to inform you that your manuscript has been judged scientifically suitable for publication and will be formally accepted for publication once it meets all outstanding technical requirements.

Kind regards,

Academic Editor

PLOS ONE

Additional Editor Comments (optional):

Reviewers' comments:

Reviewer's Responses to Questions

**Comments to the Author**

1. If the authors have adequately addressed your comments raised in a previous round of review and you feel that this manuscript is now acceptable for publication, you may indicate that here to bypass the “Comments to the Author” section, enter your conflict of interest statement in the “Confidential to Editor” section, and submit your "Accept" recommendation.

Reviewer #3: All comments have been addressed

2. Is the manuscript technically sound, and do the data support the conclusions?

Reviewer #3: Yes

3. Has the statistical analysis been performed appropriately and rigorously? 

Reviewer #3: Yes

4. Have the authors made all data underlying the findings in their manuscript fully available?

Reviewer #3: Yes

5. Is the manuscript presented in an intelligible fashion and written in standard English?

Reviewer #3: Yes

6. Review Comments to the Author

Reviewer #3: All my problems have been solved. Both the innovation and practicality meet the requirements. It is suggested that this paper be published.

7. PLOS authors have the option to publish the peer review history of their article (what does this mean? ). If published, this will include your full peer review and any attached files.

**Do you want your identity to be public for this peer review?** For information about this choice, including consent withdrawal, please see our Privacy Policy .

Reviewer #3: No

---

## [Editor Report · Acceptance letter]

PONE-D-24-50060R2

PLOS ONE

Dear Dr. ibrahim,

I'm pleased to inform you that your manuscript has been deemed suitable for publication in PLOS ONE. Congratulations! Your manuscript is now being handed over to our production team.

Kind regards,

on behalf of

Dr. Jose Santamaría López

Academic Editor

PLOS ONE